# THE CENTROID AFFINITY HYPOTHESIS: HOW DEEP NETWORKS REPRESENT FEATURES

## ABSTRACT

Understanding and identifying the features of the input a deep network (DN) extracts to form its outputs is a focal point of interpretability research, as it enables the reliable deployment of DNs. The current prevailing strategy of operating under the linear representation hypothesis (LRH) – where features are characterised by directions in a DN's latent space – is limited in its capacity to identify features relevant to the behaviour of components of the DN (e.g. a neuron or a layer). In this paper, we introduce the centroid affinity hypothesis (CAH) as a strategy through which to identify these features grounded in the behaviour of the DN's components. We theoretically develop the CAH by exploring how continuous piecewise affine DNs – such as those using the ReLU activation function – influence the geometry of regions of the input space. In particular, we show that the centroids of a DN – which are vector summarisations of the DN's Jacobians – form affine subspaces to identify features of the input space. Importantly, we can continue to utilise LRH-derived tools, such as sparse autoencoders, to study features through the CAH, along with novel CAH-derived tools. We perform an array of experiments demonstrating how interpretability under the CAH compares to interpretability under the LRH: We can obtain sparser feature dictionaries from the DINO vision transformers that perform better on downstream tasks. We can directly identify neurons in circuits of GPT2-Large. We can train probes on Llama-3.1-8B that better capture the action of generating truthful statements.

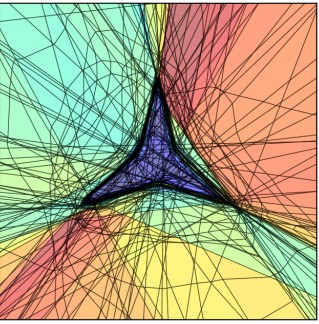 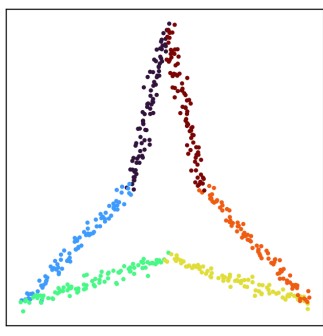 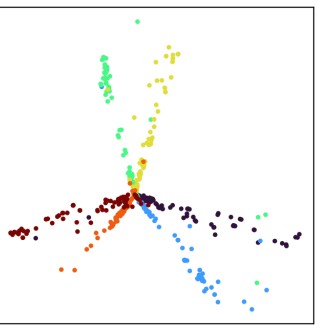

| Functional Geometry | Input Samples | Corresponding Centroids |

Figure 1: The centroid affinity hypothesis (CAH) posits that a deep network's (DN's) centroids (i.e. the sum of the rows of the Jacobians computed at the features in the input space) decompose into affine structures to identify the extracted and utilised features of the input space. This is derived by studying the functional geometry of the continuous piecewise affine (CPA) approximation of DNs, which refers to the induced partition on the input space where the CPA approximation is linear. Here we train a DN to classify the interior of a star-shaped polygon in the two-dimensional plane (see Figure 2). We sample input points from the boundary of the polygon, **centre** plot, and visualise their corresponding centroids, **right** plot, which form affine structures. The linear regions of the DN (i.e., its functional geometry) surrounding the polygon are visualised in the **left** plot using SplineCam (Humayun et al., 2023).

## 1 INTRODUCTION

Understanding and identifying what features of the input space a deep network (DN) extracts to form its output is important to facilitate the reliable deployment of DNs. Despite DNs only being constructed through a layer-wise composition of linear and nonlinear operations, this task is notoriously difficult.

The current interpretability framework for identifying these features – the linear representation hypothesis (LRH) (Elhage et al., 2022; Park et al., 2024) – abstracts away from the individual functional components of DNs (e.g. neurons and layers). This has led to a growing consensus that the LRH is limited in its capacity to provide a meaningful understanding of the role of features in the components of DNs (Sharkey et al., 2025). The LRH has encumbered the field of interpretability with contextualising features across the components of the DN (Balagansky et al., 2025) and within the DN's input space (Paulo et al., 2024).

In this paper, we propose the centroid affinity hypothesis (CAH), which inherently focuses on the relationship between features and components by studying the *centroids* of a DN. The centroids of a DN are given by the sum of the rows of the Jacobian of a component at an input point. This is in contrast to the LRH, which focuses on latent activations, which are simply the output of a component at an input point.

The CAH posits that the features of the input space that a DN uses are characterised by affine subspaces of centroids, as demonstrated in Figure 1. This characterisation is developed by studying how a DN affects the geometry of regions of the input space using the spline theory of deep learning (Balestriero & Baraniuk, 2018a). Indeed, centroids are known to parametrise the input space partition induced by the linear regions of the continuous piecewise affine (CPA) approximation of a DN (Balestriero et al., 2019).

The advantages of the CAH over other proposed frameworks for identifying architecturally-contextualised features (Bae et al., 2022; Murfet et al., 2023) are two-fold. (1) In practice, it can be considered exactly as theoretically derived, rather than requiring approximations. (2) Existing interpretability techniques, such as sparse autoencoders (Trenton Bricken et al., 2023; Huben et al., 2024), can be used to explore it.

In summary, the main contributions of our paper are as follows.

**[C1].** The centroid affinity hypothesis for how features of the input space are *extracted* and *utilised* by a DN.

**[C2].** A novel set of interpretability techniques for extracting the features of a DN and identifying their relationship to the components of the DN.

**[C3].** An illustration of how the existing interpretability techniques can be used to fruitfully explore the features of a DN using the centroid affinity hypothesis.

As a consequence of introducing the CAH, we can more effectively explore features relevant to the behaviours of components of DNs as compared to the LRH. (1) On DINO vision transformers, sparse autoencoders applied to their centroids identify a sparser feature dictionary that performs better on downstream tasks as compared to those obtained from sparse autoencoders applied to latent activations. In particular, these features persist more coherently between the DINOv2 and the DINOv3 models. (2) On GPT2, we utilise novel CAH-derived attribution metrics to corroborate prior circuit analysis work (Clement & Joseph, 2023). (3) On Llama-3.1-8B, we show how linear probes trained on centroids extract generalising features that capture the behaviour of outputting a truthful statement.

## 2 BACKGROUND

**Deep Networks and their Approximations.** A DN $f : \mathbb{R}^{d^{(0)}} \to \mathbb{R}^{d^{(L)}}$, with the convention that $d^{(0)} = d$, is a composition of $L$ functions $f^{(\ell)} : \mathbb{R}^{d^{(\ell-1)}} \to \mathbb{R}^{d^{(\ell)}}$. These functions are referred to as layers and usually constitute an affine transformation followed by a nonlinearity. Using the spline approximation theory (Lyche & Schumaker, 1975; Schumaker, 2007), DNs can be approximated to arbitrary precision or even characterised exactly using continuous piecewise affine (CPA) splines (Balestriero & Baraniuk, 2018a). More specifically, one can write $f(\mathbf{x}) \approx \mathbf{A}_{\omega_{\nu(\mathbf{x})}} \mathbf{x} + \mathbf{b}_{\omega_{\nu(\mathbf{x})}}$,

where $\mathbf{A}_{\omega_{\boldsymbol{\nu}(\mathbf{x})}} \in \mathbb{R}^{d^{(L)} \times d}$ and $\mathbf{b}_{\omega_{\boldsymbol{\nu}(\mathbf{x})}} \in \mathbb{R}^{d^{(L)}}$ are the affine parameters specific to the *linear region* $\omega_{\boldsymbol{\nu}(\mathbf{x})} \subseteq \mathbb{R}^d$ encompassing $\mathbf{x}$. Here $\boldsymbol{\nu}(\mathbf{x})$ identifies the equivalence class of $\mathbf{x}$ under the collection of all equivalence classes $\mathcal{V}$ constructed by $\sim$ where $\mathbf{x}_1 \sim \mathbf{x}_2$ if and only if $\mathbf{x}_1$ and $\mathbf{x}_2$ are in the same linear region. Similarly, approximations can be constructed for each layer $f^{(\ell)}$ to obtain

$$f(\mathbf{x}) \approx \mathbf{A}^{(L)}_{\omega_{\boldsymbol{\nu}(\mathbf{x})}^{(L)}} \left( \dots \left( \mathbf{A}^{(1)}_{\omega_{\boldsymbol{\nu}(\mathbf{x})}^{(1)}} \mathbf{x} + \mathbf{b}^{(1)}_{\omega_{\boldsymbol{\nu}(\mathbf{x})}^{(1)}} \right) \dots \right) + \mathbf{b}^{(L)}_{\omega_{\boldsymbol{\nu}(\mathbf{x})}^{(L)}} ,$$

where $\omega_{\boldsymbol{\nu}(\mathbf{x})}^{(\ell)} \subseteq \mathbb{R}^{d^{(\ell-1)}}$ denotes the linear region for the mapping $f^{(\ell)}$ encompassing $f^{(1 \leftarrow \ell-1)}(\mathbf{x}) \in \mathbb{R}^{d^{(\ell-1)}}$.[1] For CPA approximations of sub-components of the DN, say $f^{(\ell_1 \leftarrow \ell_2)} : \mathbb{R}^{d^{(\ell_1-1)}} \to \mathbb{R}^{d^{(\ell_2)}}$ for $1 \leq \ell_1 < \ell_2 \leq L$, we adopt a similar notation. When the DN employs CPA nonlinearities (e.g., ReLU), these approximations are exact (Balestriero & Baraniuk, 2018a). Henceforth, when we speak in terms of DN sub-components, we do so with the understanding that this covers everything from a single layer to the entire DN.

**Functional Geometry.** The *functional geometry* of a DN sub-component refers to the arrangement of the linear regions of its CPA approximation. That is, the functional geometry of the DN sub-component $f^{(\ell_1 \leftarrow \ell_2)}$ is the disjoint union of linear regions $\left\{ \omega_{\boldsymbol{\nu}}^{(\ell_1 \leftarrow \ell_2)} \right\}_{\boldsymbol{\nu} \in \mathcal{V}}$.

On the one hand, the linear regions forming this geometry can be thought of as being bounded by the level-sets of the nonlinearities[2] of the DN sub-component, which is a hyperplane in its input space. As the hyperplane is projected back to the input space of the sub-component, it bends at the point of intersection with the level-sets of the preceding nonlinearities (see Figure 7). It is the intersection of these planes that forms the regions which constitute the DN sub-component's functional geometry (Humayun et al., 2023).

On the other hand, the functional geometry of $f^{(\ell_1 \leftarrow \ell_2)}$ can be parametrised by a power diagram subdivision (Balestriero et al., 2019). The power diagram is a generalisation of the Voronoi tiling, which is suitable for parametrising the functional geometry of a DN. Instead of each point in the input space being assigned to the centroid it is closest to, as is the case in a Voronoi tiling, the power diagram additionally weights each centroid with a radius. That is, using a collection of centroid-radius pairs $\left\{ \left( \mu_{\boldsymbol{\nu}}^{(\ell_1 \leftarrow \ell_2)}, \tau_{\boldsymbol{\nu}}^{(\ell_1 \leftarrow \ell_2)} \right) \right\}_{\boldsymbol{\nu} \in \mathcal{V}} \subseteq \mathbb{R}^{d^{(\ell_1-1)}} \times \mathbb{R}$, each region of the partition is constructed as

$$\omega_{\boldsymbol{\nu}}^{(\ell_1 \leftarrow \ell_2)} = \left\{ \mathbf{x} \in \mathbb{R}^{d^{(\ell_1-1)}} : \boldsymbol{\nu} = \arg \min_{\boldsymbol{\nu}' \in \mathcal{V}} \left( \left\| \mathbf{x} - \mu_{\boldsymbol{\nu}'}^{(\ell_1 \leftarrow \ell_2)} \right\|_2^2 - \tau_{\boldsymbol{\nu}'}^{(\ell_1 \leftarrow \ell_2)} \right) \right\}.$$

**Proposition 2.1** (Balestriero et al. 2019). *Let* $\mathbf{J}_{\mathbf{x}} \left( f^{(\ell_1 \leftarrow \ell_2)} \right) \in \mathbb{R}^{d^{(\ell_1-1)} \times d^{(\ell_2)}}$ *denote the Jacobian of* $f^{(\ell_1 \leftarrow \ell_2)}$ *at* $f^{(1 \leftarrow \ell_1-1)}(\mathbf{x})$. *Then,* $\mu_{\boldsymbol{\nu}(\mathbf{x})}^{(\ell_1 \leftarrow \ell_2)} = \left( \mathbf{J}_{\mathbf{x}} \left( f^{(\ell_1 \leftarrow \ell_2)} \right) \right)^{\top} \mathbf{1}$.

From Proposition 2.1, it follows that the parameterisation of the functional geometry of a DN is computationally accessible through Jacobian-vector products.

**Feature and Circuits.** Regions of the input space of a DN consist of a combination of *atomic* features that are useful for the DN to generate its output. For example, an image of a cat probably represents the atomic features of whiskers and a furry tail. Therefore, to extract an atomic feature (e.g., whiskers), the DN must encode their *superposition* for regions of the input space and aggregate them to delineate them. For the atomic feature to influence the output of the DN, this delineation should be such that the DN can use it to influence the behaviour of its components. For example, inducing a particular activation pattern in its nonlinearities, or triggering an attention head. The precise influence that an atomic feature has on the computational graph of a DN is often termed a *circuit*. With this, the task of interpretability is to understand how to identify these features of the

---

[1]It should be understood that in this context, $\boldsymbol{\nu}(\mathbf{x})$ identifies the equivalence class on the linear regions in $\mathbb{R}^{d^{(\ell-1)}}$.

[2]By level-set, we refer to the points in space – whether that be in the input space of the DN or the input space of the nonlinearity – that activate the nonlinearity at its knots. For example, for the ReLU nonlinearity, the level set would refer to the points that are zero when input into the nonlinearity.

input space that the DN has *extracted* to *influence* its output. Henceforth, we will usually refer to these as the features of the DN.

The linear representation hypothesis (LRH) posits that these features emerge as linear directions within the latent spaces of a DN. Evidence for the LRH was first teased out using text semantics in word-embedding models (Mikolov et al., 2013), before being supported in transformers (Nanda et al., 2023). This simple perspective facilitated the development of a variety of computationally efficient tools to perform feature extraction (Kim et al., 2018; Trenton Bricken et al., 2023; Huben et al., 2024) and understand the behaviours of DNs (Liu et al., 2019; Hewitt & Manning, 2019; Templeton et al., 2024; Arditi et al., 2024; Lin et al., 2024). Furthermore, the LRH is amenable to theoretical characterisation, allowing for the systematic study of DN features (Arora et al., 2016; Park et al., 2024; 2025). However, since the LRH abstracts away from the individual components of the DN, there is a growing consensus that the LRH does not reliably identify features relevant to the computational graph of a DN (Sharkey et al., 2025). That is, the LRH may identify features that have been *extracted* but not *utilised* by the DN, as opposed to *functionally relevant* features.

**Definition 2.2.** A functionally relevant feature of a DN sub-component is an extracted feature of the input space that is utilised by particular components of the DN. With the exact nature of this utilisation referred to as the corresponding circuit.

In Appendix B, we explain how this description of features and circuits is analogous to prior work.

## 3 THE CENTROID AFFINITY HYPOTHESIS

In Section 3.1, we theoretically motivate and derive the centroid affinity hypothesis (CAH). In Section 3.2, we support the CAH with a simple experiment of a DN classifying whether a two-dimensional input point is inside or outside a star-shaped polygon.

### 3.1 DERIVING THE CENTROID AFFINITY HYPOTHESIS

**Our Hypothesis.** Humayun et al. (2024) characterised the grokking phenomenon in DNs (Power et al., 2022) – the delayed generalisation of DNs – as the migration of linear regions from the training data to the decision boundary. Due to the inherent connection between linear regions and the nonlinearities of the DN, this was compared with the phenomenon of circuit clean-up (Nanda et al., 2022) – the observation that redundant circuits are discarded as the DN generalises. In particular, Humayun et al. (2024) demonstrated that this region migration phenomenon – which results in linear regions aligning in the input space – is a *universal* phenomenon of DN training dynamics, with connections to the generalisation and robustness of the DN. Similarly, the functional geometry of a DN sub-component is known to characterise its properties, such as toxicity (Balestriero et al., 2023) and (Cosentino & Shekkizhar, 2024) reasoning in large language models.

Thus, it is intuitive to then expect linear regions to be pertinent to identifying the functionally relevant features of a DN. Indeed, this amounts to our central hypothesis, which we introduce now.

**Definition 3.1** (*Informal*). Functionally relevant features of a DN sub-component are represented by collections of *aligned* linear regions in the input space. (Formalised in Appendix C).

**Features as Aligned Nonlinearities.** The nonlinearities of the DN sub-component form these regions through their level-sets, which are hyperplanes within the input space of the nonlinearities. Meaning nonlinearities can only align in the input space of the DN sub-component when the latent activations form a linearly identifiable boundary within the input space of the nonlinearity. Therefore, we have that *the functionally relevant features of the $\ell^{th}$ layer of a DN sub-component are the regions of its input space whose latent activations are linearly identifiable within the input space of the $\ell^{th}$ layer and populated with the level-sets of the nonlinearities of the $\ell^{th}$ layer.*

**Features as Affine Centroids.** For practical purposes, we can convert this characterisation of features as aligned nonlinearities into the power diagram perspective.

**Proposition 3.2** (*Informal*). *The functionally relevant features of the $\ell^{th}$ layer of a DN sub-component are represented by centroids that form an affine subspace in $\mathbb{R}^{d^{(\ell-1)}}$. (Formalised and Proved in Appendix C).*

To pull back the features of the $\ell^{\text{th}}$ layer into the input space of the larger sub-component, it suffices to iteratively take the intersection with the regions of the previous layers of the sub-component. Using Proposition 2.1, this is done by performing a linear projection based on the activation pattern of the points in the region at the previous layers of the sub-component. This refinement maintains the affine structure of the original centroids, although now a single feature at the $\ell^{\text{th}}$ layer may partition into multiple features in the input space of the sub-component. This neatly describes how a DN sub-component can iteratively refine the superposition of features in its input space into atomic features that it can utilise to form its output. We formalise this reasoning with Theorem 3.3.

**Theorem 3.3** (*Informal*). *The functionally relevant features of a DN sub-component are represented by the affine structures in its corresponding centroids. (Formalised and Proved in Appendix C).*

In light of Theorem 3.3, we characterise our hypothesis on how the functionally relevant features of a DN are represented as the *centroid affinity hypothesis* (CAH).

**Connection to the LRH.**   The CAH offers a subtle, although complementary, difference from the LRH. The LRH requires that the latent activations of features form affine structures, which is a stronger condition than being linearly identifiable. Moreover, the way the LRH is utilised and interpreted implies that any set of activations that form affine structures corresponds to features of the DN (Smith, 2024). However, in the CAH, we require both the linear identifiability of latent activations and that the level-sets of the nonlinearities separate them. For if no nonlinearities separate them, then moving along these affine structures would not induce a distinct difference in the behaviour of the DN, and thus not represent a functionally relevant feature (see Definition 2.2).

**Centroid Stability Identifies Circuits.**   Circuits are identified by quantifying the relationship between components of a DN and features through attribution methods (Meng et al., 2022; Wang et al., 2023; Goldowsky-Dill et al., 2023). In light of Theorem 3.3, we propose to use the sensitivity of centroids to manipulations in the components of a DN as an attribution method.

Formally, let $f$ be a DN and $f^{(i,\ell)}$ be the same DN but with the $i^{\text{th}}$ neuron of the $\ell^{\text{th}}$ manipulated. The attribution of neuron $i$ to the features of a collection of samples $\mathcal{N}$ is quantified as

$$s_{\mathcal{N}}^{(i,\ell)} := \frac{1}{|\mathcal{N}|} \sum_{\mathbf{x} \in \mathcal{N}} \frac{\left\| \mu_{\mathbf{x}}^f - \mu_{\mathbf{x}}^{f^{(i,\ell)}} \right\|_2}{\left\| \mu_{\mathbf{x}}^f \right\|_2}, \tag{1}$$

where $\mu_{\mathbf{x}}^f$ and $\mu_{\mathbf{x}}^{f^{(i,\ell)}}$ are the centroids of $f$ and $f^{(i,\ell)}$ at $\mathbf{x}$ respectively. Quantifying the attribution of a neuron to the local features of a sample point $\mathbf{x}$ can be done by taking $\mathcal{N}$ to be $\mathcal{B}_\epsilon(\mathbf{x}) = \left\{ \mathbf{x}' \in \mathbb{R}^d : \| \mathbf{x} - \mathbf{x}' \|_2 < \epsilon \right\}.$[3]

### 3.2   Supporting the Centroid Affinity Hypothesis

Here we consider the validity of the CAH for a DN trained to classify whether two-dimensional input points are inside or outside the star-shaped polygon of Figure 2 (far left). In this instance, the atomic features of the input space – that are relevant for the task of the DN – are the interior and exterior of the polygon. In the input space, these exist as a superposition of other features, such as the edges and vertices of the polygon.

**Centroid Affinity for Feature Identification.**   In Figure 2, we observe that the nonlinearities of the last hidden layer of this DN align themselves along the boundary of the polygon, which identifies the interior and exterior as a functionally relevant feature of the DN. Consequently, with the far right plot of Figure 2, we support Proposition 3.2, since the centroids of the interior and exterior of the polygon form affine subspaces in the input space of the last layer.

Moreover, with the right plot of Figure 2, we support the idea that the CAH requires linear separability between the latent activations. Additionally, with the right plot of Figure 2, we observe that the latent activations of the exterior form three separate directions. Under the LRH, these would correspond to three distinct features; however, since no nonlinearities align in these regions, they will not relate to

---

[3]Henceforth, we will use $s^{(i,\ell)}$ to denote $s_{\mathcal{B}_\epsilon(\mathbf{x})}^{(i,\ell)}$ unless stated otherwise.

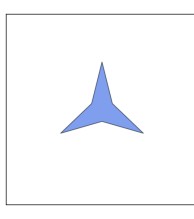 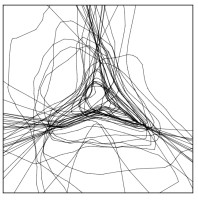 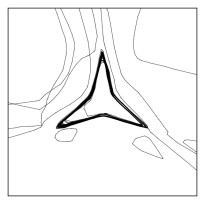 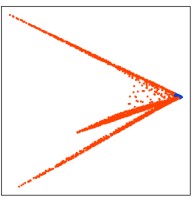 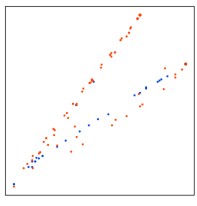

Input Polygon  $2^{\text{nd}}$ Hidden Layer  $3^{\text{rd}}$ Hidden Layer  Activations  Centroids

Figure 2: Here we train a fully connected ReLU DN with three hidden layers to classify the interior of a star-shaped polygon in two-dimensions, **far left**. We visualise the functional geometry of the second and the third hidden layer components with the **left** and **centre** plots, respectively. In the **right** and **far right** plots, we project the latent activations and centroids obtained from the third hidden layer onto their first two principal components, respectively. Orange points correspond to input samples from outside of the polygon, and blue points correspond to input samples from inside the polygon.

the components of the DN. Consequently, performing interpretability under the LRH in this instance would lead to identifying functionally irrelevant features, as we demonstrate in Appendix D.

Similarly, the second hidden layer identifies the three edges of the polygon as features, as the nonlinearities align along these boundaries (evidenced by the extending hyperplanes emanating from each of the three tips of the polygon). Thus, with Figure 1 we support Theorem 3.3 since the centroids of the linear regions bounding the edges of the polygon form affine subspaces segmented according to which edge they identify.

This demonstrates how the DN uses linear regions, and thus its nonlinearities, to extract the superposition of features in the input space into atomic features that can then be used to form its outputs. In particular, this process yields coherent structures in the centroids of the DN as claimed by the CAH.

**Point-Cloud Analysis.**  Under the LRH, a standard interpretability technique is studying point clouds of latent activations using embedding methods (Li et al., 2025), topological descriptors (Fay et al., 2025) or linear probes (Kim et al., 2018; Nanda et al., 2023). In addition to centroids possessing an affine structure under the CAH, it is evident from Figure 1 that this structure is semantically coherent. Suggesting that applications of these analyses may be particularly fruitful under the CAH.

To aid the study of centroids as point clouds, we consider softening DNs using CPA nonlinearities (e.g., ReLU), by replacing these nonlinearities with smooth approximations (e.g., GELU). We justify this in Appendix F.

First, we consider directly measuring the distribution of centroid affinity, using a notion of effective dimension,[4] to identify feature boundaries. In Figure 3, we see that this identifies the external sectors and interior of the polygon as features of the input space.

Second, we consider a t-SNE embedding (van der Maaten & Hinton, 2008) of the centroids at the second hidden layer obtained from the samples of Figure 8. In Figure 3, we observe a neat partition with respect to which sector of the input domain the input samples were located. This corroborates our prior analysis using the second hidden layer's functional geometry.

We reproduce these findings for DNs trained on other polygons in Appendix G and extend the analyses to a convolutional neural network trained on MNIST (Lecun et al., 1998) in Appendix H.

**Circuit Discovery.**  Using Equation (1), we can compute neuron attribution values to study the circuits of this DN by observing the sensitivity of neurons to pruning. In Figure 3 (right), we see that the neurons of the third hidden layer have an encompassing effect on the entire boundary of the polygon – as expected from our prior analyses.

---

[4]Here we measure effective dimension using the exponential of the entropy of the normalised singular values.

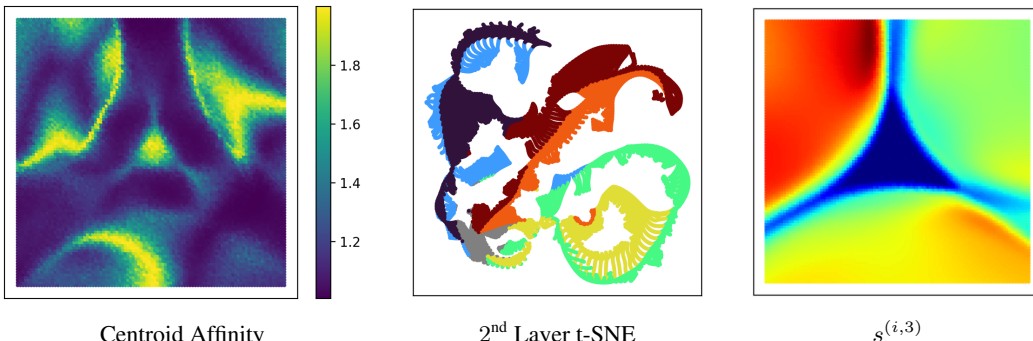

| Centroid Affinity | $2^{\text{nd}}$ Layer t-SNE | $s^{(i,3)}$ |

Figure 3: Here we study the centroids of the softened DN of Figure 2. In the **left** plot, we take individual points in the input space, obtain a $128$ sized sample of points within a $0.4$ radius of this point, and compute the effective dimension of the corresponding centroids. In the **centre** plot, we took the input sample of Figure 10 and embedded – using t-SNE – their corresponding centroids obtained at the second hidden layer. Points within the polygon are coloured grey, whilst the other points are coloured depending on what sector of the input space they came from. In the **right**, we visualise the influence of neurons from the third hidden layer on the centroids of points sampled in the input space, as given by Equation (1).

## 4    EXPERIMENTS

We now use the CAH to interpret larger DNs, including the vision transformers DINOv2 (Oquab et al., 2024) and DINOv3 (Siméoni et al., 2025), GPT2 (Radford et al., 2019) and Llama-3.1-8B (Grattafiori et al., 2024). In Appendix L, we detail the differences in computational resources required to perform these experiments under the CAH compared to under the LRH.

### 4.1    FEATURE EXTRACTION WITH SPARSE AUTOENCODERS

Here, we explore the features of sparse autoencoders trained on the latent activations and centroids from DINO models (Oquab et al., 2024; Siméoni et al., 2025). Similar to Hindupur et al. (2025), we extract the latent activations of Imagenette (Jeremy Howard, 2025) from these models to train a TopK sparse autoencoder (Gao et al., 2025) – in Appendix I we explore the robustness of the learned dictionaries of these sparse autoencoders when applied to the full ImageNet dataset (Krizhevsky et al., 2012). However, we additionally consider training a TopK sparse autoencoder on the centroids extracted from the last multi-layer perceptron block of the models.

**Generalising, Sparse and Functionally Relevant Features.**    First, we compare the feature dictionaries extracted from a DINOv2 model (Oquab et al., 2024) in the following ways:[5] Train linear probes on the feature decompositions of the train set of Imagenette to classify its classes, and then evaluate the accuracy of the probe on the feature decompositions of the test set of Imagenette. Record the frequency at which the features of the sparse autoencoder fire on the test set of Imagenette. Record the activation pattern similarity ratios of an input sample.

The activation pattern similarity ratio is computed as follows: We sample an input point and record the Jaccard similarity between its feature decomposition and the feature decomposition of the other points in the train set. We then compute the Jaccard similarity of its binarised latent activation in the DINOv2 model to the other points in the train set. The activation pattern similarity ratios are then these latter quantities divided by the former pairwise.

In Figure 4, we observe that the feature dictionaries from the centroid sparse autoencoders yield linear probes with higher accuracy and a more uniform firing distribution on the test set of Imagenette. Moreover, the activation pattern similarity ratios are generally larger, which means similar feature decompositions correspond to similar activation patterns in the DINOv2 model.

---

[5]We additionally provide a more qualitative analysis in Appendix J.

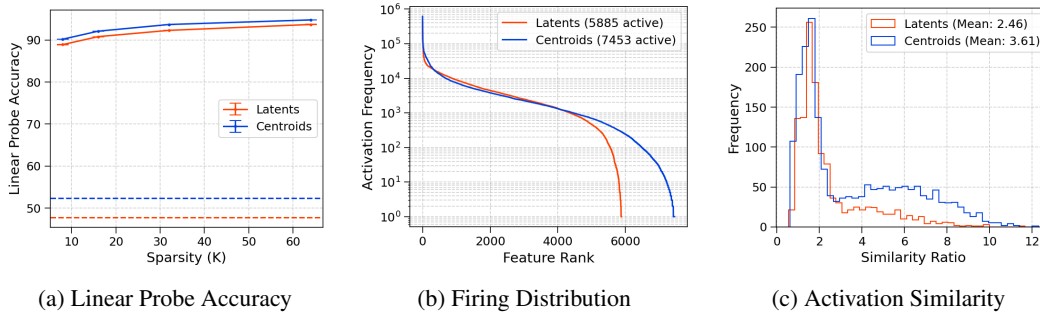

(a) Linear Probe Accuracy      (b) Firing Distribution      (c) Activation Similarity

Figure 4: For the **left** and **centre** plots, we train sparse autoencoders on the latent activations and centroids extracted from all the tokens of the DINOv2 feature extractor applied to the Imagenette train dataset. An expansion factor of $10$ is used with sparsity values in the range $\{8, 16, 32, 64\}$. In the **left** plot, we measure the accuracy of the linear probes, and in the **centre** plot, we measure the firing distribution of the 32-K sparse autoencoder. For the linear probes we consider five random initialisations and we report a baseline accuracy obtained by applying PCA reduction to the latent activations and centroids directly, rather than applying a sparse autoencoder. In the **right** plot, we train a 32-K TopK sparse autoencoder with an expansion factor of $10$ on the latent activations and centroids extracted from the class token of the DINOv2 feature extractor. We then record the activation similarity ratios for an input sample.

**Persistent Features.** Next, we compare the feature dictionaries of the sparse autoencoders when applied to the DINOv2 model and the DINOv3 (Siméoni et al., 2025) model. Intuitively, we would expect DINOv3 to refine the features learned by DINOv2. Consequently, the cosine similarities of feature dictionaries of the corresponding sparse autoencoders should have a large mass concentrated around one with a long tail toward lower values.[6]

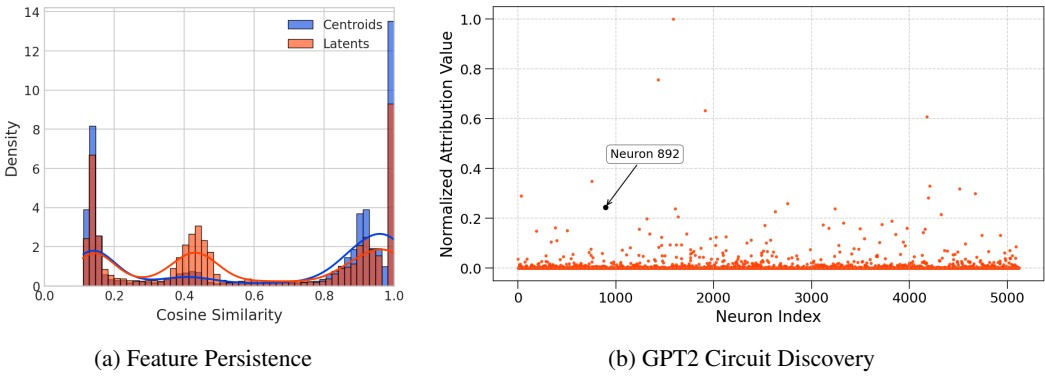

(a) Feature Persistence      (b) GPT2 Circuit Discovery

Figure 5: In the **left** plot, we record the pairwise cosine similarities between the dictionary features of centroid and latent activation TopK sparse autoencoders obtained from DINOv2 and DINOv3. In the **right** plot, we prompt GPT2-Large and note the normalised attribution value (using Equation (1)) in a neighbourhood of the embedding at the input of the multi-layer perceptron block at the thirty-first layer of the last token of this prompt. The neighbourhood is constructed by sampling 256 points within a radius of $0.25$ of the embedding. We normalise these values to be between zero and one. In black we indicate the $892^{\text{nd}}$ neuron in the multi-layer perceptron.

Indeed, this is precisely what we observe for the sparse autoencoders trained with centroids, see Figure 5a. Whereas the sparse autoencoders trained with latents have a bimodal distribution, where some features have a high cosine similarity and others have a cosine similarity of around $0.4$. This bimodal distribution is concerning as it suggests the identified features do not correlate between the models, and are instead just spurious artefacts.

---

[6]We may additionally observe lots of features with low cosine similarity, since the learned dictionaries have inactive features.

## 4.2 CIRCUIT DISCOVERY

In Clement & Joseph (2023), it was observed that GPT2-Large has a neuron in the multi-layer perceptron component of the thirty-first layer, which is responsible for predicting the "an" token. This was determined by observing the effects of latents on the model. It was concluded that this neuron works in concert with other neurons within the multi-layer perceptron block to capture the corresponding feature. We support this by computing neuron attribution values with Equation (1) using a neighbourhood of the last token embeddings at the input of the thirty-first layer on the prompt "I climbed up the pear tree and picked a pear. I climbed up the apple tree and picked". In Figure 5b, the distribution of neuron attribution values is heavily skewed, with the neuron identified by Clement & Joseph (2023), marked in black, sitting within the top $99.8^{th}$ percentile of values. In Appendix K, we use this example to explore the robustness of Equation (1) as an attribution metric.

## 4.3 PROBING

Probing is another technique that exploits linear structures in DNs to either extract features (Kim et al., 2018), extract representations (Nanda et al., 2023), or classify inputs (Marks & Tegmark, 2024). Here, we explore the latter of these applications by forming linear classifiers to discern the truthfulness of input statements to large language models. More specifically, we adopt the mass-mean probes of Marks & Tegmark (2024) along with their dataset to test the generalisation capacities of probes. Using the `likely` dataset, we obtain mass-mean probes from the twelfth layer of the Llama-3.1-8B large language model (Grattafiori et al., 2024), either using the latent activations or the centroids extracted from the multi-layer perceptron component. The `likely` dataset is formed

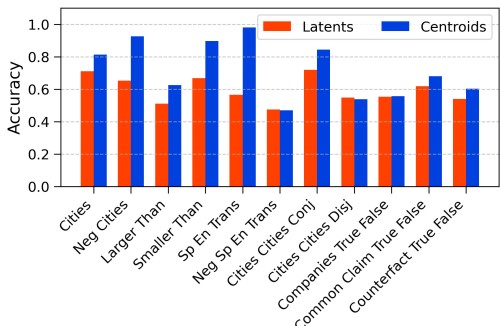

Figure 6: We obtain mass-mean probes on the `likely` dataset from the twelfth layer of Llama-3.1-8B, either using latent activations or centroids extracted from the multi-layer perceptron component. We then test these probes using a collection of other datasets from Marks & Tegmark (2024).

as a classification problem on whether sample tokens are likely or unlikely under the model's logit distribution from non-factual textual inputs. The other datasets are formed as a classification problem between factually truthful or untruthful statements. Therefore, `likely` is identifying plausibility in the model's outputs rather than a concept of truthfulness. Consequently, by virtue of the fact that centroid-based mass-mean probes generalise more effectively to the truth-identifying datasets (see Figure 6), it follows that centroids capture the action of outputting a truthful statement rather than the concept of a truthful statement.

## 5 DISCUSSION

In this work, we have attempted to address the limitation of the LRH of not being aware of the components of the DN. To do so, we appealed to the spline theory of deep learning to explore how the components of the DN affect its functional geometry, where the functional geometry of a DN refers to the arrangement of the linear regions of its continuous piecewise affine approximation. From this perspective, we proposed the *centroid affinity hypothesis* (CAH), which posits that the functionally relevant features of a DN are represented as affine structures in the DN's centroids.

Under the CAH we can continue to leverage interpretability techniques derived under the LRH. Indeed, we applied sparse autoencoders to centroids to extract sparser feature dictionaries from vision transformers that are robust and perform strongly on downstream tasks. Furthermore, we showed that centroids identify features relevant to the behaviour of DNs by obtaining probes from the centroids of Llama-3.1-8B that generalise better than probes obtained from latent activations. The CAH also introduces novel interpretability tools, which we use to perform circuit discovery on GPT2.

In summary, the CAH provides a complementary perspective to the LRH for identifying DN features in a manner that is less susceptible to spurious or over-fitted features. A limitation of CAH currently is that it cannot be directly utilised for feature steering, unlike the LRH.

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

## A    MOTIVATING EXAMPLE

In Section 3.2, we utilised a simple example to demonstrate how the CAH can be used to identify features in practice. In Figure 7, we use this example to indicate how the nonlinearities of a DN form its geometry.

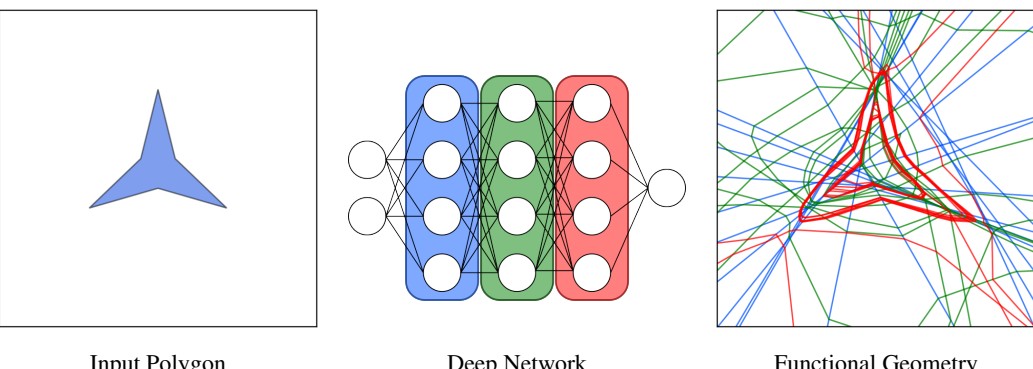

Input Polygon                    Deep Network                    Functional Geometry

Figure 7: A DN has a functional geometry formed by its nonlinearities. Each nonlinearity identifies a boundary within the input space of the DN. These boundaries start off as hyperplanes in the input space of the nonlinearity. They are then projected back to the input space of the DN by intersecting with the hyperplanes of the nonlinearities of the previous layer. These then bound the linear regions which constitute the functional geometry of the DN. Consequently, the blue nonlinearities appear as hyperplanes in the input space, the level-sets of the green nonlinearities only bend when they intersect with the level-sets of the blue nonlinearities, etc. On the **left**, we visualise the underlying data distribution that the DN is being trained on. In the **centre**, we visualise a simplified schematic of the architecture of the DN. In this schematic, we highlight the nonlinearities in the first, second and third hidden layers of the DN; each of which constructs a hyperplane within the input space of the DN, which we identify in the **right** plot. The **right** plot depicts the functional geometry of the DN (using SplineCam (Humayun et al., 2023)), having trained on the polygon of the **left** plot.

## B    COMPARISON TO PRIOR WORK

**Features.**    Definition 2.2 is analogous to the notion of a DN concept used in Park et al. (2024), which provides a rigorous theoretical characterisation of the LRH. In particular, in Park et al. (2024), a DN concept is a variable that leads to a particular output when caused by a context. In Definition 2.2, the context would correspond to the regions of the input space that poses a given feature. The variable notion of Park et al. (2024) would then be equivalent to the *extraction* requirement of Definition 2.2, since the assumed distribution on the input space would define a distribution over these regions. Similarly, the referenced causation on the output would be equivalent to the *utilisation* requirement of Definition 2.2.

**Circuits.**    Circuits were initially introduced in Olah et al. (2020) to deal with the apparent poly-semantic nature of neurons. That is, specific neurons were observed to trigger on seemingly semanti-cally disjoint inputs, whereas ensembles of neurons demonstrated more reliable activation patterns. Instead, our notion of a circuit arises naturally as the responses of the components of a DN to a feature. In particular, this response is likely to incorporate multiple neurons or components of a DN due to the DN's compositional construction.

## C    FORMAL THEORY

**Definition C.1.**  A feature of the $\ell^{\text{th}}$ layer of a DN is a collection of regions constructed by hyperplanes whose normals have a pair-wise cosine similarity bounded below by $1 - \epsilon$, and whose closest points to the origin have a pair-wise Euclidean distance bounded above by $\delta$.

**Proposition C.2.** *The functionally relevant features in the input space of the $\ell^{th}$ layer of a DN sub-component are represented by centroids that form an affine subspace in $\mathbb{R}^{d^{(\ell-1)}}$ with deviations proportional to approximately $\sqrt{2\epsilon}$.*

*Proof.* Let the corresponding centroids and radii of the $\ell^{th}$ of the DN be $\left\{ \left( \mu_{\boldsymbol{\nu}}^{(\ell)}, \tau_{\boldsymbol{\nu}}^{(\ell)} \right) \right\} \subseteq \mathbb{R}^{d^{(\ell-1)}} \times \mathbb{R}$. Then suppose $\Pi$ and $\tilde{\Pi}$ are hyperplanes forming the feature. Each hyperplane, say $\Pi$, corresponds to a boundary of two regions, such that

$$\Pi = \left\{ \mathbf{x} : \left\| \mu_1^{(\ell)} - \mathbf{x} \right\|_2^2 - \tau_1^{(\ell)} = \left\| \mu_2^{(\ell)} - \mathbf{x} \right\|_2^2 - \tau_2^{(\ell)} \right\}$$
$$= \left\{ \mathbf{x} : \left\langle \mu_1^{(\ell)} - \mu_2^{(\ell)}, \mathbf{x} \right\rangle = c \right\},$$

for some constant $c$. Thus, $\Pi$ has normal vector $\mathbf{n}_1 := \mu_1^{(\ell)} - \mu_2^{(\ell)}$. Similarly, we can assume $\tilde{\Pi}$ is such that it has normal $\mathbf{n}_2 := \mu_2^{(\ell)} - \mu_3^{(\ell)}$. By assumption, we have that

$$\frac{\mathbf{n}_1 \cdot \mathbf{n}_2}{\|\mathbf{n}_1\|_2 \|\mathbf{n}_2\|_2} \geq 1 - \epsilon.$$

Thus, by using small angle approximations, the angle between the normal vectors $\theta$ is approximately less than $\sqrt{2\epsilon}$. In particular, $\mathbf{n}_2$ can be decomposed into components parallel and orthogonal to $\mathbf{n}_1$ as

$$\mathbf{n}_2 = \|\mathbf{n}_2\|_2 \cos(\theta) \hat{\mathbf{n}}_1 + \|\mathbf{n}_2\|_2 \sin(\theta) \hat{\mathbf{u}},$$

where $\hat{\mathbf{n}}_1$ is the unit vector of $\mathbf{n}_1$ and $\hat{\mathbf{u}}$ is normal to it. Consequently, we can write

$$\begin{cases} \mu_1^{(\ell)} = \mu_1^{(\ell)} + 0 \cdot \mathbf{d} \\ \mu_2^{(\ell)} = \mu_1^{(\ell)} - \|\mathbf{n}_1\|_2 \mathbf{d} \\ \mu_3^{(\ell)} = \mu_2^{(\ell)} - \|\mathbf{n}_2\|_2 \cos(\theta) \mathbf{d} - \|\mathbf{n}_2\|_2 \sin(\theta) \hat{\mathbf{u}}, \end{cases}$$

where $\mathbf{d} = \hat{\mathbf{n}}_1$. Therefore, since $\sin(\theta)$ is of order $\sqrt{2\epsilon}$. Repeating this for each tuple of three regions, we conclude that all the centroids forming a feature lie in the same affine subspace. Thus, the proof is complete. $\square$

**Theorem C.3.** *The features of a DN sub-component are represented by centroids that form approximate affine subspaces.*

*Proof.* Suppose that the feature corresponds to a $(\epsilon, \delta)$-feature of the $\ell^{th}$ layer. Then by Proposition C.2, the centroids at the $\ell^{th}$ form an approximate affine subspace. Thus, for sufficiently small $\delta$, the DN sub-component corresponds to an affine transformer, meaning the corresponding centroids within the input space of the DN sub-component also form an approximate affine subspace. $\square$

## D   RE-EVALUATING INTERPRETABILITY TOOLS

An advantage of the CAH over other frameworks for identifying functionally relevant features is that interpretability tools derived under the LRH can be utilised. For example, sparse autoencoders (Trenton Bricken et al., 2023; Huben et al., 2024) for constructing dictionaries of features and transcoders (Dunefsky et al., 2024) for circuit extraction.

Here, we demonstrate that sparse autoencoders applied to latent activations of the DN of Section 3.2 are insufficient for identifying the interior of the polygon as a feature, and spuriously assign multiple directions to the exterior of the polygon. However, sparse autoencoders applied to centroids only identify the two directions corresponding to the interior and exterior of the polygon as features. Likewise, transcoders are insufficient at reconstructing the functionally relevant features of the DN.

**Sparse Autoencoders.** Sparse autoencoders are a method for extracting an over-complete basis for a set of vectors (Trenton Bricken et al., 2023; Huben et al., 2024), with the aim of identifying *meaningful* directions. A sparse autoencoder has an architecture of the form $g(\mathbf{z}) = \mathbf{W}_{\text{dec}}\sigma\left(\mathbf{W}_{\text{enc}}\mathbf{z} + \mathbf{b}_{\text{enc}}\right)$, with $\mathbf{W}_{\text{enc}} \in \mathbb{R}^{d^{\text{feat}} \times d^{\text{act}}}$, $\mathbf{b}_{\text{enc}} \in \mathbb{R}^{d^{\text{feat}}}$, $\mathbf{W}_{\text{dec}} \in \mathbb{R}^{d^{\text{act}} \times d^{\text{feat}}}$, where $d^{\text{act}}$ is the dimension of the set of vectors and $d^{\text{feat}}$ is the size of the over-complete basis which is to be constructed. In the context of a DN, a sparse autoencoder is trained to reconstruct its latent activations with an added sparsity regularisation term. The idea is that the rows of $\mathbf{W}_{\text{dec}}$ would then constitute a dictionary of features, with the term $\sigma\left(\mathbf{W}_{\text{enc}}\mathbf{z} + \mathbf{b}_{\text{enc}}\right)$ giving the decomposition of the activation $\mathbf{z}$ in terms of these features.

Under the CAH, it is not necessarily the case that the latent activations of features correspond to linear directions. This means that sparse autoencoders may not extract every feature, and may extract spurious features instead. For example, in Figure 8, the sparse autoencoder trained on latent activations identifies directions that are not functionally relevant (see Figure 2). Another problem with this approach is the neglect of any functional information.

These problems can be mitigated by applying sparse autoencoders to reconstruct centroids rather than latent activations. In Figure 8, we see that sparse autoencoders trained on centroids only recover the affine structures of the centroids that correspond to the features of the DN.

**Transcoders.** Transcoders have a similar architecture to sparse autoencoders, except they are trained to reconstruct the input-output mapping of a DN sub-component rather than its activations (Dunefsky et al., 2024). Although this approach is more faithful to the *function* of the DN sub-component, it is inherently limited since the transcoder only has one hidden layer and so its functional geometry is not very expressive. In Figure Figure 8, we see that the transcoder's functional geometry is not faithful to that of the DN sub-component, meaning it has not captured its underlying features.

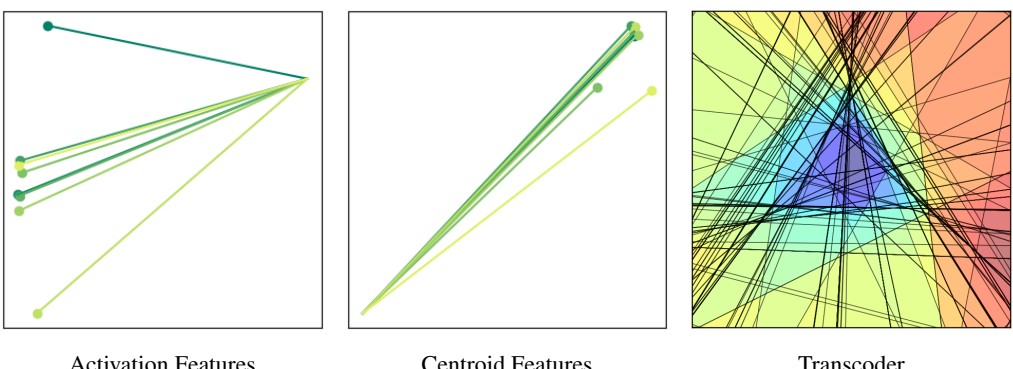

| Activation Features | Centroid Features | Transcoder |

Figure 8: Sparse autoencoders identify the meaningful affine structures present in DN centroids, and are susceptible to identifying linear structures in latent activations that have no functional relevance to the DN. Transcoders are limited in their capacity to reconstruct the features of a DN. In the **left** and **centre** plots, we train a sparse autoencoder to reconstruct the latent activations and softened centroids of the input sample from Figure 10 at the last hidden layer of the DN, respectively. The projections of these latent activations and centroids can be observed in Figure 2. In the **right** plot, we train a transcoder with twice as many features as neurons in the hidden layer of the DN whose functional geometry is visualised in the left plot of Figure 1.

# E  EMERGENCE OF CENTROID STRUCTURE

The centroid affinity of Figure 1 emerges gradually through training. At initialisation, the centroids have a similar arrangement to the input samples, due to the random initialisation of the DN. However, as training progresses, we observe that the centroids slowly migrate and align themselves. In particular, we can see the alignment of the centroids manifest before they arrive at their ultimate position.

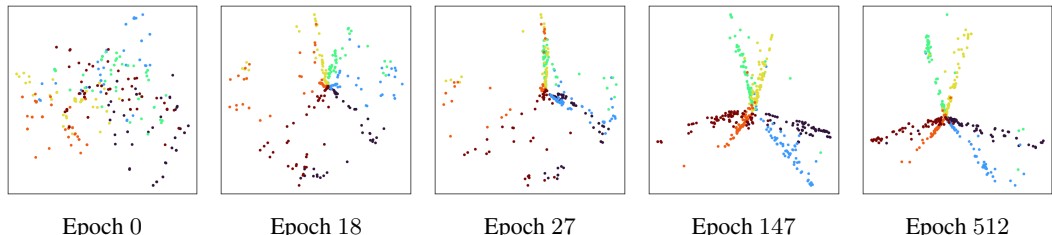

| Epoch 0 | Epoch 18 | Epoch 27 | Epoch 147 | Epoch 512 |

Figure 9: Throughout the training of the DN of Figure 1, we tracked the DN centroids of the input samples highlighted in Figure 1.

## F  SOFTENING DEEP NETWORKS

We developed the CAH by studying the level-sets of nonlinearities, which is a property of CPA DNs (i.e., those implementing CPA nonlinearities, like ReLU). We argued that this was valid since any DN can be approximated to arbitrary precision by CPA DNs. However, for these DNs, the centroids are discrete objects since they exist uniquely for each linear region, which may present a challenge since Theorem 3.3 is a necessarily continuous utilisation of centroids. Therefore, here we consider the effect of using smooth nonlinearities on the CAH.

Firstly, to allow for better analyses of CPA DNs, we will explore the effect of relaxing their non-linearities to smooth nonlinearities. For example, for the DN of Section 3, we consider *softening* it by replacing the ReLU nonlinearities with GELU nonlinearities (Hendrycks & Gimpel, 2023). The GELU nonlinearity belongs to the swish family of nonlinearities (Ramachandran et al., 2017), which are theoretically known to provide an appropriate softening of a ReLU DN's functional geometry (Balestriero & Baraniuk, 2018b). In Figure 10, we see that by softening the DN, we maintain and add more detail to the structure of the centroids.

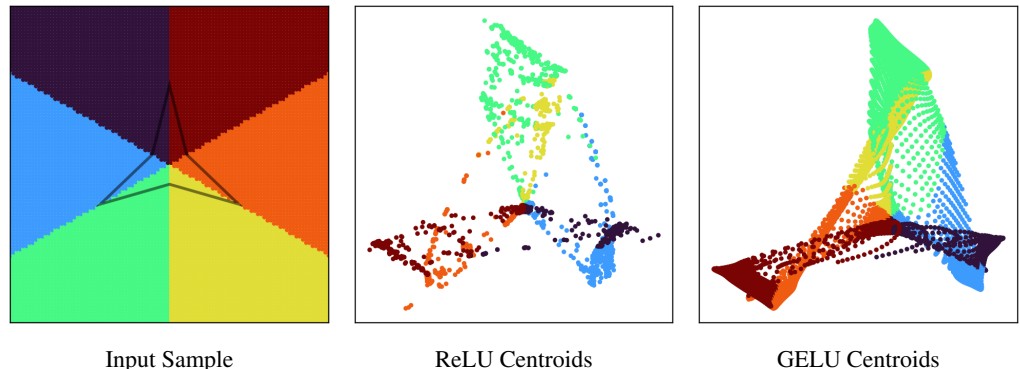

| Input Sample | ReLU Centroids | GELU Centroids |

Figure 10: Softening a DN with ReLU nonlinearities by replacing them with GELU nonlinearities provides more detail to the structure of the centroids without affecting their overall structure. Here we sample a grid of points in the input space of the polygon-classifying DN of Figure 1, **left** plot, and compute their corresponding centroids when the ReLU nonlinearity is maintained, **centre** plot, and when the ReLU nonlinearity is replaced by the GELU nonlinearity, **right** plot.

Secondly, we determine whether our investigations of Appendix D hold for DNs trained from scratch using continuous nonlinearities. That is, for a DN with GELU nonlinearities, we perform the exact same training procedure for the DN considered in Section 3, and then analyse the resulting centroids.

From Figure 11 we observe similar features as those identified for the ReLU DN considered in Appendix D: when replacing the nonlinearity back to a ReLU we can observe its functional geometry using SplineCam (Humayun et al., 2023) and we see the alignment of the nonlinearities around the polygon, when we observe the centroids of the input samples from Figure 10 we see the same affine structures that arose in the ReLU DN, computing centroid affinity for points in the input space again

identifies the edges of the polygon as a feature, the influence of pruning neurons on the centroids is still effective as a neuron attribution metric.

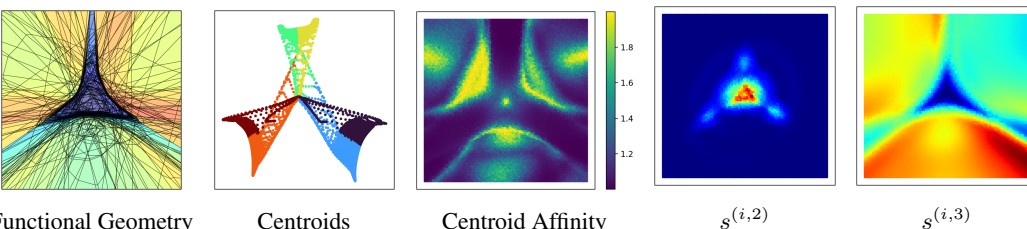

Functional Geometry    Centroids    Centroid Affinity    $s^{(i,2)}$    $s^{(i,3)}$

Figure 11: Here we train a DN in the same manner as the one considered in Figures 1 and 2, except we use the GELU nonlinearity. In the **far left** plot, we replace the nonlinearities with ReLU such that we can use SplineCam to visualise its functional geometry. In the **left** plot, we visualise the centroids from the input samples of Figure 10. In the **centre** plot, we compute the centroid affinity of points in the input space based on a sample of radius $0.4$. In the **right** and **far right** plots, we consider the sensitivities of centroids when pruning neurons from the second and third layers, respectively.

## G    OTHER POLYGONS

In addition to the star-shaped polygon considered in the main text, in Figure 12 we corroborate the observed patterns when the input distribution is a bowtie-shaped and reuleaux-shaped polygon.

## H    MNIST CENTROID STRUCTURE

Thus far, we have seen theoretically and in a simple example how the centroids of a DN have a semantically coherent structure. Here, we demonstrate how this can be used to explore the *feature boundaries* of a DN trained on the MNIST classification task (Lecun et al., 1998). For this, we train a DN with a convolutional feature extractor followed by a linear layer on MNIST. After training, we sample two inputs from distinct classes and compute centroid affinity values – at the feature extractor component of the DN – along the linear interpolation between the samples. We observe in Figure 13 that there is a greater relative drop in centroid affinity between more distinct classes. More specifically, the 3 and 6 classes are intuitively more distinct than the 4 and 9 classes; consequently, centroid affinity is lower along the interpolation between the 3 and 6 inputs since the features are more distinct. Whereas, if we similarly consider the effective dimensions of the latent activations, we do not observe any contextual change.

## I    IMAGENET FEATURE STRUCTURE

In addition to the Imagenette dataset considered in Figure 4, we also consider TopK sparse autoencoders trained on ImageNet (Krizhevsky et al., 2012). More specifically, we train TopK sparse autoencoders on the class token extracted from the DINOv2 feature extractor, with the intention of understanding how robust the CAH framework is to hyperparameters. We train such sparse autoencoders across multiple random initialisations, with varying sparsity parameters and expansion factors. We compare the resulting feature dictionaries using the Centred Kernel Alignment metric (Kornblith et al., 2019).

From Figure 14, we observe that the dictionaries obtained from centroids are more similar across random initialisations. In particular, this holds consistently as the sparsity parameter and expansion factors are varied.

## J    QUALITATIVE ANALYSIS OF FEATURES

In Section 4, we demonstrated quantitatively that the features extracted from a sparse autoencoder trained to reconstruct centroids were semantically- and functionally-relevant to the input distribution

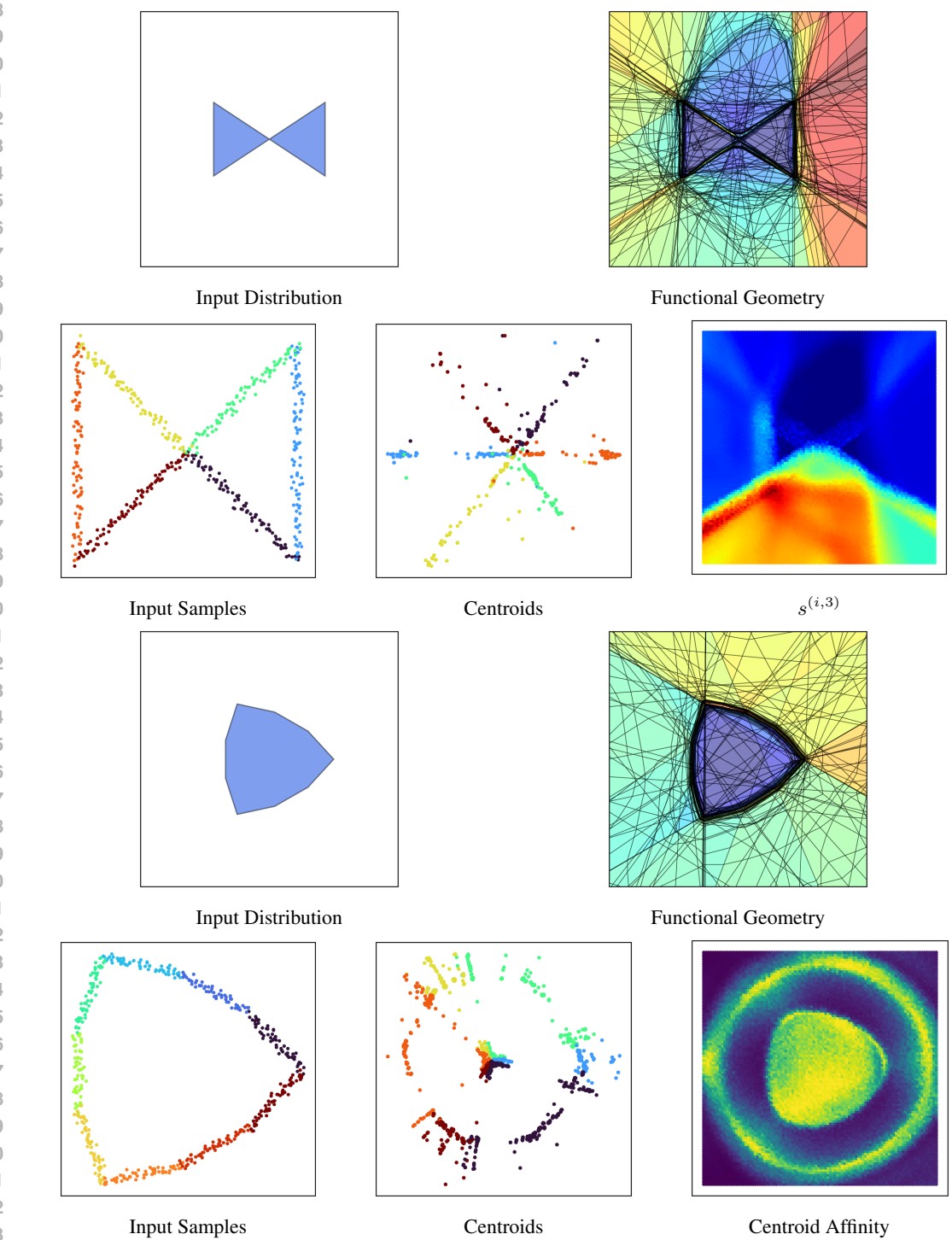

Figure 12: Here we perform some of the same analyses as conducted previously, but with a DN trained on a bowtie-shape polygon, **top** two rows, and a reuleaux-shaped polygon, **bottom** two rows.

and feature extractor. Here, we qualitatively support this and compare them to the features extracted by the sparse autoencoder trained to reconstruct latent activations. To do so, we randomly sample a point from the input distribution and identify the other inputs from the distribution with similar feature decompositions – as measured by Jaccard similarity.

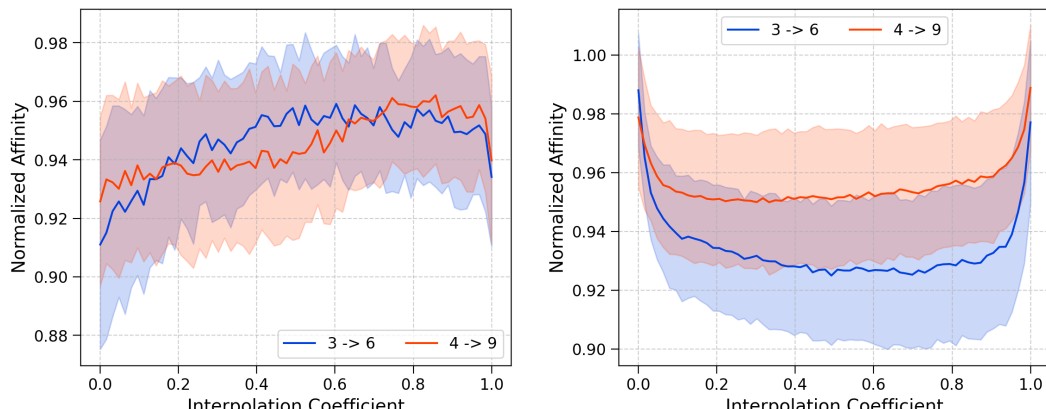

Figure 13: We train a DN on the MNIST classification. In the **right** plot, we consider the CA of centroids at the feature extractor component of the DN – which constitutes three convolutional layers. In the **left** plot, we similarly consider the effective dimensions of the latent activations. For two training points of distinct classes, we compute the CA of samples along their linear interpolation. We visualise these affinities for samples from class pairs (3,6) and (4,9).

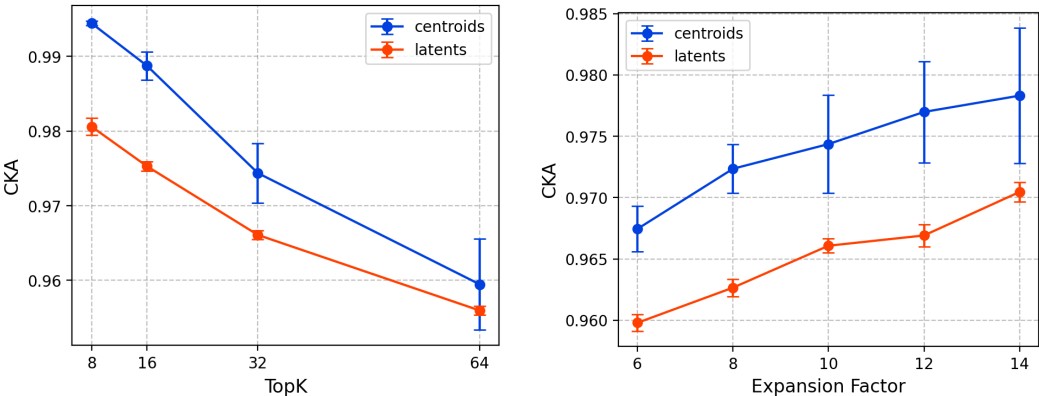

Figure 14: The dictionaries learned by sparse autoencoders trained on centroids are more similar, as formalised by the CKA metric, across random initialisations as compared to the dictionaries of sparse autoencoders trained on latent activations. Here we train sparse autoencoders on the class token of the DINOv2 extractor when applied to ImageNet. We repeat this over five random initialisations, and then we compare the dictionaries of these sparse autoencoders using the CKA metric. Reported are the mean values along with one standard deviation. In the **left** plot we consider how these values change as the sparsity parameter of the sparse autoencoder is changed. In the **right** plot we consider how these values change as the expansion factor of the sparse autoencoder is changed.

In Figure 15, we see that the sparse autoencoder trained to reconstruct centroids identifies similar features to those of the sparse autoencoder trained to reconstruct latent activations. This further supports that the centroids of a DN have a coherent structure that can be used to identify the features of the DN.

## K    ROBUSTNESS OF NEURON ATTRIBUTION WITH CENTROIDS

In order for Equation (1) to prove useful as a tool for interpreting DNs, it is essential that it is robust in its application. For example, Equation (1) ought not be sensitive to the neighbourhood $\mathcal{N}$ chosen. Furthermore, since in practice we can only approximate Equation (1) by taking a finite sample of points from $\mathcal{N}$, it is important that Equation (1) has low variance in relation to this finite sample. In Figure 16 we test both these properties for the experiment of Figure 5b. In the left plot, we

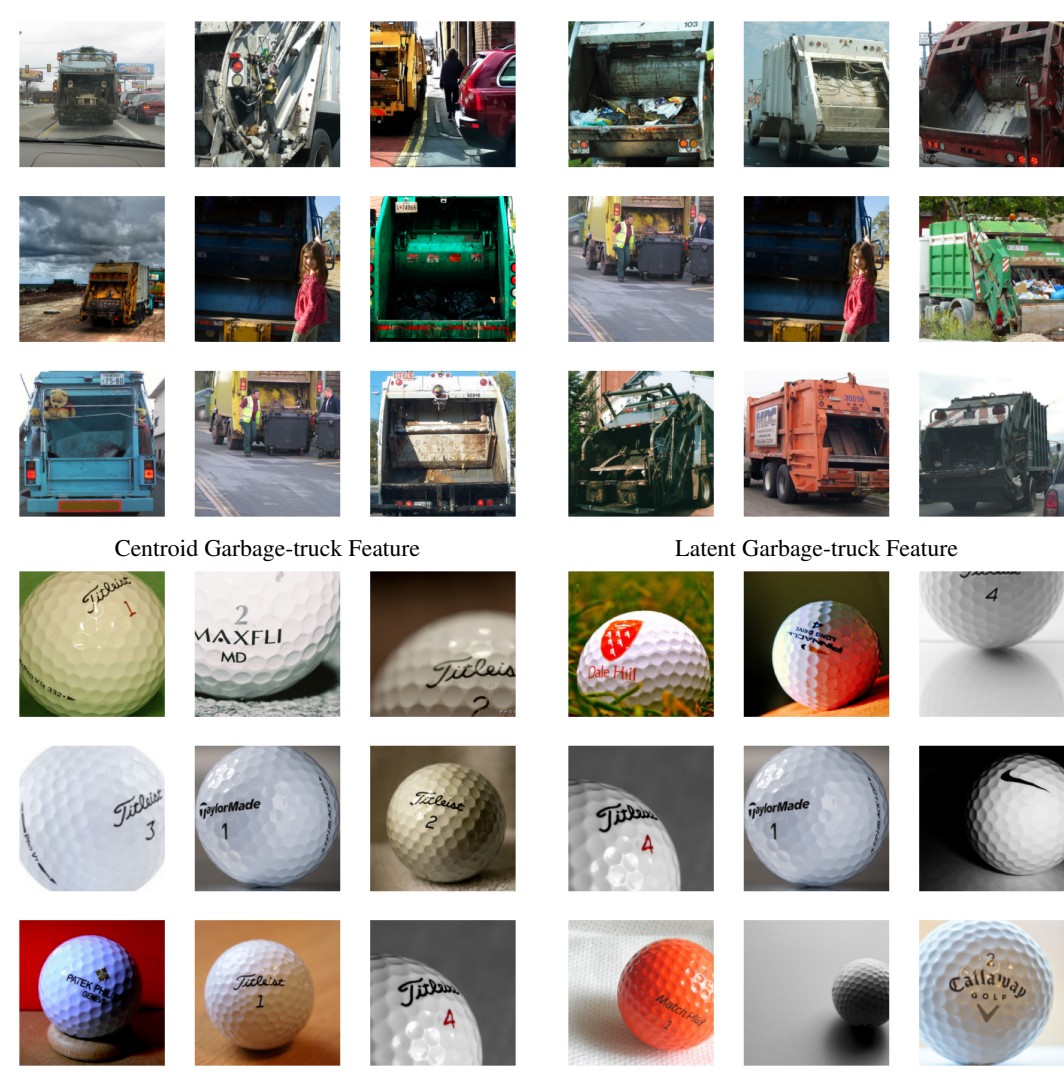

Centroid Garbage-truck Feature       Latent Garbage-truck Feature

Centroid Golf-ball Feature       Latent Golf-ball Feature

Figure 15: For a sampled input, we compute its feature decomposition using the sparse autoencoders of Figure 4c, and then identify the other inputs whose feature decompositions are most similar to this using Jaccard similarity. More specifically, in the **left** column, we consider the sparse autoencoder of Figure 4c trained using centroids, and in the **right** column, we consider the sparse autoencoder of Figure 4c trained using latent activations. The central image of each plot represents the initial point that is sampled from the input distribution, and the surrounding images are the identified inputs with similar feature decompositions. Each **row** considers a specific input.

observe that attribution values have a low variance when a finite sample is used to approximate the neighbourhood $\mathcal{N}$. In the right plot we observe that the percentile of a particular neuron within the layer of the DN is stable across different neighbourhood sizes. This ensures that conclusions derived from Equation (1) are robust.

## L    COMPUTATIONAL REQUIREMENTS.

A valid concern with exploring the CAH is the computational burden it introduces into the process of interpretability, since it requires interrogating the Jacobians of a DN. Fortunately, this interrogation only requires considering Jacobian vector products (see Proposition 2.1), which are significantly cheaper to compute in common computational frameworks. Furthermore, often the analysis of

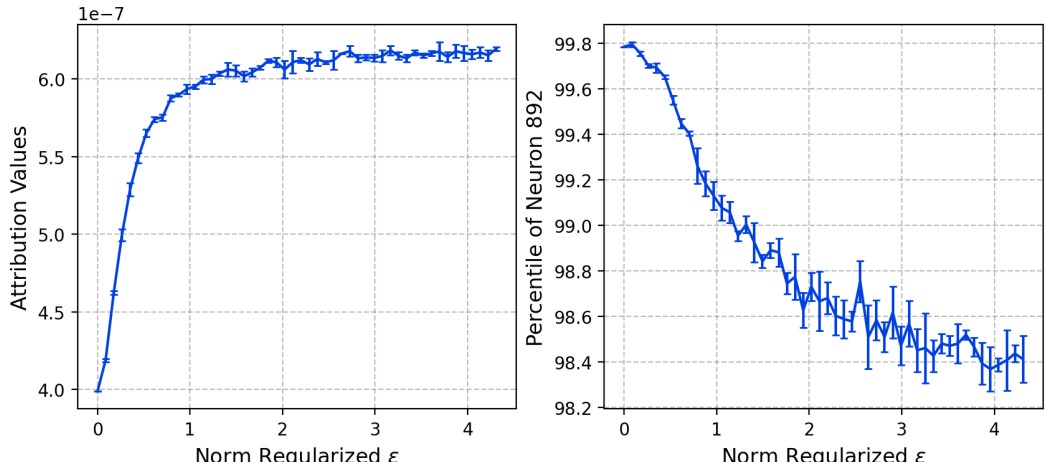

Figure 16: The neuron attribution metric of Equation (1) is a robust measure for interpreting the neurons of a DN. Here we consider the robustness of Equation (1) for the experiment in Figure 5b. More specifically, we test how the attribution values of the neurons changes as we consider increasingly large neighbourhoods. The neighbourhoods we consider are of the form $\mathcal{B}_\epsilon(\mathbf{x})$, where $\mathbf{x}$ is the embedding of the last token of a prompt at the $31^{\text{st}}$ of GPT2-Large. We consider $\epsilon$ normalized by the norm of the centroid of $\mathbf{x}$ at this layer of the DN. To compute Equation (1) at each neuron of the layer, we sample 256 embeddings from this neighbourhood. In the left plot we observe how the average attribution value of each neuron changes across random samplings of this neighbourhood. In the right plot we observe how the percentile value of the $892^{\text{nd}}$ changes across these random samplings. The error bars represent one standard deviation in the observed values.

centroids is concentrated on a relatively small component of the DN. Thus, this computation would appear relatively insignificant compared to processing the entire DN. In this section, we empirically quantify the computational burden incurred by considering centroids rather than latents in our main experiments of Section 4.

**DINO Feature Extraction.** For this experiment, we compare the difference in computational time to extract the latent activations and centroids from the feature model. After these vectors are extracted, the computational pipeline is identical when using centroids or latent activations. We summarise the results in Table 1, where we see that extracting centroids only increases the computation time by around $10\%$. Especially since this extraction phase only amounts to a small proportion of the entire experiment, this difference is almost negligible.

**GPT2 Circuit Discovery.** Although there is no direct analogue of this experiment with latent activations, we can still argue that the computational burden is relatively benign. In particular, since we only compute centroids across the multi-layer perceptron block of the thirty-first layer, we only need to consider the Jacobian vector product for this component. This can be done by storing gradients of a forward pass across this block, which, in relation to performing a forward pass across the model, is insignificant.

**Llama-3.1-8B Probes.** As in the DINO feature extraction experiments, the only difference in computation times is in extracting the centroids from the model. Table 1 shows that this difference is relatively larger for centroids. However, in the grand scheme of saving the activations and loading the model, this difference is still relatively minor.

**MNIST Centroid Structure.** For this experiment, we compare the time necessary to perform the experiment with centroids or latent activations. More specifically, instead of considering the effective dimensions of the centroids of neighbourhoods of points, we compute the effective dimensions of the latent activations of neighbourhoods of points. We summarise the results in Table 1, where we see that using centroids instead of latent activations requires $14\%$ more computation time.

Table 1: Here, we compare the computation times (in seconds) for using the centroids of a DN to perform interpretability to using latent activations.

| Experiment | With Centroids | With Latent Activations |
|---|---|---|
| DINO Feature Extraction (Figure 4) | 8.7 | 7.8 |
| Llama-3.1-8B Probes (Figure 6) | 2329 | 470 |
| MNIST Centroid Structure (Figure 13) | 881 | 771 |

