# OpenReview forum: "The Centroid Affinity Hypothesis: How Deep Network Represent Features"
_ICLR.cc/2026/Conference — Submitted to ICLR 2026_

### Official Review · Reviewer_gL46 · 2025-10-31

**Soundness:** 2
**Presentation:** 2
**Contribution:** 1
**Rating:** 2
**Confidence:** 3

**Summary:**

This paper proposes the Centroid Affinity Hypothesis (CAH) as an alternative framework to the Linear Representation Hypothesis (LRH) for identifying features in deep neural networks. The core claim is that functionally relevant features, which are actually used by network components rather than merely represented, can be identified through centroids (sums of Jacobian rows) that form affine subspaces. The theoretical foundation is based on spline theory and the functional geometry of continuous piecewise affine (CPA) networks, suggesting that aligned linear regions correspond to features that the network actively utilizes. The authors validate CAH through experiments on toy problems, DINO vision transformers, GPT-2, and Llama-3.1-8B, claiming improvements in feature sparsity, generalization, and circuit discovery over LRH-based methods.

**Strengths:**

Strengths:

1. The limitation that LRH abstracts away from individual network components is legitimate and widely recognized in the interpretability community. The distinction between "extracted" versus "utilized" features is conceptually valuable.

2.  The grounding in spline theory and functional geometry is rigorous.

**Weaknesses:**

Weaknesses:

1.  Figure 1 Lacks Critical Methodological Details: The "Functional Geometry" visualization using SplineCam doesn't specify which layer, which hyperplanes are shown. While Section 3.2 suggests these are real experimental results (not illustrations), the artistic rendering and lack of detail make reproducibility and understanding difficult.

2. Section 2 is notation-heavy, which makes it difficult to follow. Motivation for functional geometry is lacking and requires detailed discussion.

3. The main paper provides an informal theorem to show that the functionally relevant features of a deep network sub-component are represented by the affine structures in its corresponding centroids and defers the formal theorem and proof to the appendix.

4. Proposition 3.2 (Formal Proof for Theorem 3.3) shows three specific centroids $(\mu_1, \mu_2, \mu_3)$ have a geometric relationship where $\mu_3$ deviates from the line through $\mu_1,\mu_2$ by an amount proportional to $\sqrt{2\epsilon}$. To say centroids form "an affine subspace," authors need to show that there exists a consistent affine coordinate system where ALL centroids in the feature can be expressed. The last equation in the formal proof has a sign issue. Using the provided information, we get
\begin{equation} \mu^{(l)}_3=  \mu^{(l)}_2   + || n_2||_2 \cos(\theta)d + ||n_2||_2 \sin(\theta) \hat{u}, \end{equation}
But the paper has the opposite sign for the second and third terms. Please clarify.

**Questions:**

Questions to Authors:
1. In Figure 1, for the "Functional Geometry" plot, which layer's geometry is visualized? What are the SplineCam visualization parameters? Are all linear regions shown or a subset?

2. Power diagram formulation lacks motivation and needs discussion on why this parameterization is required.

3. What happens with different neighborhood sizes ($\epsilon$ in $B_{\epsilon(x)}$)?

4. Please answer the questions in the weaknesses section.

---

> ### Author Response · Authors · 2025-11-18
> **Concerns Regarding Methodological and Theoretical Details**
>
> We appreciate the detailed review of our paper, particularly on its theoretical aspects, which rigorously ground interpretability. We would like to address your questions below and acknowledge the subsequent corrections we have made to strengthen the rigour of our paper.
>
> **Figure 1 lacks critical methodological detail.**
>
> The SplineCam visualisation is of how the hyperplanes of the deep network (across each of its layers) intersect the two-dimensional input space. They are not artistic renderings, but rather represent the empirical results of training a deep network to classify the interior and exterior of a polygon in a two-dimensional plane. We acknowledge that this can be made more explicit in the text, and so we have rectified that in an updated version.
>
> **Section 2 is notation-heavy, and the formal proofs are deferred to the appendix.**
>
> We agree that Section 2 utilises a lot of notation; however, we find it necessary to build up the CAH. Therefore, to improve the accessibility of our work, we made the decision to defer the formal statements of the theorems to the appendix to prevent having to introduce further notation.
>
> **The functional geometry/power diagram formulation lacks motivation.**
>
> The functional geometry and its power diagram parametrisations are certainly conceptually challenging objects. In hindsight, we could have motivated them more to demonstrate why the challenge of comprehending them is fruitful. The functional geometry of a CPA deep network is just an artefact of it having the CPA property. It is a partitioning of the input space into regions on which the deep network reduces to a specific affine transformation. This partitioning is inherently related to the architecture and parameters of the deep network. This is why it is of interest for understanding the properties of the deep network. Indeed, in prior works (Humayun et al. 2024), (Balestriero et al., 2023) and (Cosentino & Shekkizhar, 2024), this functional geometry has been related to the robustness of deep networks, and the toxicity and reasoning of large language models (these are referenced in the "Our Hypothesis" paragraph starting on line 189 of the updated paper).
> However, understanding the functional geometry from the perspective of intersecting hyperplanes is combinatorial and thus intractable.
> Another way to partition an input space is as a Voronoi tiling, and this is precisely what KNNs do – they set regions based on the proximity of input points to a collection of vectors. Although we cannot reconstruct the functional geometry of a deep network as a Voronoi tiling, a generalisation of the Voronoi tiling (i.e., the power diagram) can be parametrised to reconstruct the functional geometry of a deep network. The centroids are precisely the parameters of the power diagram that reconstruct the functional geometry of a deep network. In the updated paper, we elaborate on this connection to make in lines 134-137.
>
> **Queries regarding the mathematical derivation of Proposition 3.2 and a typo.**
>
> We appreciate the time you have taken to go through the details of our theoretical statements and derivations.
> Firstly, you are correct regarding the typo; we have a sign error. Thank you for pointing this out; however, it does not affect the validity of the proof.
> Secondly, regarding the proof itself, by Def C1, we are assuming that a feature corresponds to hyperplanes whose normals are approximately cosine-similar. In Proposition C2, we are considering one such feature. By choosing three neighbouring regions, we show that their centroids form a line. Since we can do this for consecutive tuples of three regions, it follows that all the centroids representing that feature form one affine subspace. We acknowledge that we did not explicitly state this last step; therefore, we added it to the updated version.
>
> **How do the neighbourhood sizes affect our attribution metric?**
>
> This is a very interesting question. Firstly, it is essential that the neuron attribution metric is relatively stable to ensure its reliable application in practice. Secondly, how the attribution metric varies across scales can offer insights into how the functional geometries, and thus features of the deep network, vary at different scales. For our GPT2 experiment, we found that the percentile of the 892nd neuron amongst all neurons in that layer was stable across a range of neighbourhood sizes. As the neighbourhood sizes become large, the percentile value starts to drop, which is expected as the neighbourhood sampling becomes more sparse. Furthermore, the metric is stable with regard to the samples within a particular neighbourhood. In the updated paper, we have included Appendix K to provide further details on this topic.

---

### Official Review · Reviewer_LpT5 · 2025-11-01

**Soundness:** 3
**Presentation:** 2
**Contribution:** 3
**Rating:** 6
**Confidence:** 2

**Summary:**

The paper proposes a new method to investigate how deep nets represent features. The key contribution is Centroid affinity hypothesis (CAH), which states that the features of the input space that a DN uses are characterised by affine subspaces of centroids. This is different than the traditional, and widely-used Linear representation hypothesis (LRH) which focuses on latent activations.

The paper demonstrates how this CAH makes sense by using an example of polygon interior classification. The paper also demonstrates that using CAH as feature representations instead of latent activations can achieve higher probing accuracy in some experiments.

CAH can also be used to estimate the attribution of a neuron to the features of a collection of samples. To that end, the paper demonstrates how using CAH to identify a neuron for article 'an' in GPT-2.

**Strengths:**

* This paper introduces a novel and interesting hypothesis (CAH) for how features are represented in deep networks. The paper points out the differences between CAH and the traditional LRH.

* The experimental results are good to show that centroids are more useful than latent attentions for probing. Especially, the experiment that CAH can be used to identify the neuron for article 'an' in GPT2 is interesting and seems helpful for circuit discovery on LLMs.

* Although the technical details are difficult to follow, the implementation is simple and and thus it's quite straightforward to replicate the paper.

**Weaknesses:**

* The content can be challenging to understand. Some places can be made clearer. For instance, it would be helpful to state clearly that the example in section 3.2 (fig 2) is about classifying whether a pixel is in-or-outside the polygon. Also, it's unclear why the centroids in fig2 (right most) is different than the centroids in fig 1 (right most)

* The paper focuses only on ReLU. How about other activation functions?

* The experiment results on probing accuracy may be not very convincing for why one should use CAH. Could simple noise reduction (e.g. applying PCA to latent activations) also work?

**Questions:**

Please see the weaknesses.

---

> ### Author Response · Authors · 2025-11-18
> **Concerns Regarding Clarity, Generalisation to Beyond ReLU Activation Functions, Probing Results**
>
> Thank you for the thoughtful review of our paper and for highlighting how, despite the CAH having a simple implementation, the subsequent empirical results are promising. We are happy to address your questions below and incorporate the subsequent changes to improve the clarity of our paper.
>
> **The content can be challenging to understand.**
>
> Thank you for providing this feedback. It is clear that, upon re-reading our paper, improvements can be made to enhance its clarity. For example, we have been more explicit in the construction of the example of Section 3.2. Similarly, we provide a greater motivation and introduction to the power diagram concept in lines 134-137.
> The centroids of Figure 2 are obtained from the third hidden layer of the deep network, as compared to the full deep network in Figure 1. In the caption of Figure 2, we have been more explicit about this.
>
> **The paper focuses only on the ReLU activation function.**
>
> Indeed, in the main text, we motivate and derive the CAH with deep networks using the ReLU nonlinearity. Since it is in this setting that the centroid as an object is motivated as parametrising the functional geometry. However, the centroid itself is computable for arbitrary deep networks. In our experiments of Section 5, the deep networks use the GELU nonlinearity. In Appendix F, we provide concrete evidence that relaxing the use of ReLU in deep networks is justifiable.
>
> **Could simple noise reduction also work, in relation to probing?**
>
> This is certainly a valid question, and thus we consider this in our updated set of experiments. In the left plot of Figure 4, we add baseline results obtained from probes trained on a PCA decomposition of the latent activations and centroids. We find that the baselines perform significantly worse; however, the centroid baseline outperforms the latent activation baseline.

---

### Official Review · Reviewer_QquJ · 2025-11-02

**Soundness:** 3
**Presentation:** 2
**Contribution:** 2
**Rating:** 4
**Confidence:** 3

**Summary:**

This paper introduces the Centroid Affinity Hypothesis (CAH), a theoretical framework towards the analysis of the representation of features in deep networks. The core idea revolves around "centroids", i.e., vector summarisations of the Jacobians of the network. Unlike the Linear Representation Hypothesis (LRH), which assumes that features correspond to directions in latent space, CAH aims to connect features directly to the network’s functional geometry and its nonlinear components.

**Strengths:**

The paper offers a compelling geometric reinterpretation of feature representation in neural networks and provides a well motivated framework that connects spline-based geometry with interpretability.

**Weaknesses:**

My main concerns revolve around the experimental/empirical evaluation of the approach and any potential insights that might stem from it.

*The empirical evaluation lacks depth*: The experiments primarily illustrate the CAH qualitatively and focus on reporting differences (e.g., higher sparsity, slightly better probe performance) without offering substantial interpretive insight. There is no rigorous statistical or diagnostic analysis to confirm that the CAH leads to more faithful or causally relevant features. Can the authors provide clearer empirical validation or quantitative tests of faithfulness and functional relevance?

Despite strong theoretical framing, the experiments do not yield new mechanistic understanding of models. The DINO and GPT2 results largely restate known patterns or replicate existing findings under the new representation. What unique interpretability or mechanistic insight does CAH enable that previous frameworks could not?

The linear probe and sparse autoencoder tests rely on standard benchmarks but lack baselines, error bars, or ablation results. There is no clarity on how results generalize across runs or hyperparameter choices. Could the authors clarify whether CAH performance differences persist under varying conditions? What happens when the full ImageNet is considered instead of the small Imagenette?

**Questions:**

Please see the Weaknesses section.

---

> ### Author Response · Authors · 2025-11-18
> **Concerns Regarding Empirical Validation**
>
> Thank you for the detailed review of our paper and for acknowledging that CAH is a well-motivated framework for enhancing interpretability. Below, we answer your comments on its empirical aspects, which, in turn, improve the practical support for CAH.
>
> **The experiments primarily illustrate CAH qualitatively and focus on reporting differences…without offering substantial interpretive insights.**
>
> This is a fair point; our experiments are primarily focused on comparing the CAH to the LRH. On the one hand, this was one of our main intentions, as the derivation of the CAH was motivated by the need to overcome some of the limitations of the LRH – limitations that we outline in the paper. On the other hand, we provide results that generate insights beyond the LRH. For example, on Llama 3.1-8B, we discover directions that are responsible for outputting truthful statements – something that cannot be deduced from the LRH.
>
>
> **There is no rigorous statistical or diagnostic analysis to confirm that the CAH leads to more faithful or causally relevant features.**
>
> This is a valid point; our lack of providing statistical observations on the difference between CAH and LRH limits the clarity of our results. To remedy this, we repeated our linear probe experiments across five random initialisations and additionally reported the standard deviations of these results (see left of Figure 4).
> Furthermore, we trained sparse autoencoders across different random initialisations and compared the resulting dictionaries using the centred kernel alignment (CKA) metric. We observed that across random initialisations, the dictionaries learned by the SAEs trained on centroids are more similar. This provides additional evidence that CAH is more robust than the LRH (see Appendix I).
>
> **The experiments do not yield new mechanistic understanding of models.**
>
> We appreciate this concern, and our response to the first part mitigates this, as it identifies the conclusions from our Llama 3.1-8B as new. However, our experiments also demonstrate that the CAH is a more robust hypothesis for considering the features of a deep network as compared to the LRH. For example, the persistence of dictionaries across models within a family (i.e., Figure 5a) opens up the opportunity to reliably perform interpretability across models within model families. This is an avenue of future research that we are excited about.
>
> **The linear probe and sparse autoencoder tests rely on standard baselines but lack baselines, error bars and ablation results.**
>
> As mentioned above, we remedy this by training linear probes across random initialisations. Furthermore, we supply a baseline which involves reducing the latent activations or centroids to 256 dimensions through PCA (explaining 90% of the variance) and training linear probes on these features. We find that the linear probes perform significantly worse, although the probes trained on the centroids perform better.
> Although we considered the effect of the sparsity parameter K in the original paper, we additionally studied the dictionaries as the expansion factor of the SAE is varied. Notably, for this experiment, we utilised ImageNet (see Appendix I). For practical tractability, we couldn’t consider the SAEs trained on each token patch but rather only the class token. However, it represents an applicability of CAH beyond small datasets. In these experiments, we found that the dictionaries learned by the sparse autoencoders when trained on centroids were more similar.

---

### Author Response · Authors · 2025-11-18
**Paper Update and Reviewer Feedback**

Dear Reviewers,

Thank you for the time and effort you put into providing detailed reviews of our paper. We have provided an updated paper based on your comments, to which we have responded below. In summary, we enhanced the empirical validation of the CAH by incorporating baselines, ablations, statistical errors, and experiments that extend to the ImageNet dataset. We improved the clarity of exposition and corrected some minor errors. To be explicit, we used blue text for our updates.

We are happy to answer any further questions you may have.

Many thanks, Authors

---

### Meta-Review · Area_Chair_Axwn · 2025-12-21

**Summary:**

This paper has received mixed review scores (4, 6, and 2). The major concerns include insufficient quantitative analysis of the method, limited methodological insight and generalization ability, and the absence of a clearly articulated motivation.

**Reviewer Concerns:**

In the rebuttal, the authors clarified their motivation and provided additional details, which may help address concerns regarding readability and motivation. However, the rebuttal does not include further quantitative analysis of the method, nor additional experiments demonstrating its generalization ability.

**Reviewer Scores:**

Since the reviewer who assigned a score of 6 expressed limited confidence in his rating, he may lower their score due to the unresolved concerns regarding quantitative evaluation and generalization raised in the comments. For the other two reviewers, their ratings are likely to remain unchanged, as the concerns about methodological analysis and generalization persist.

---

### Decision · Program_Chairs · 2026-01-26

Reject